# Sero-surveillance for IgG to SARS-CoV-2 at antenatal care clinics in three Kenyan referral hospitals: Repeated cross-sectional surveys 2020–21

Ruth K. Lucinde[1]*, Daisy Mugo[1], Christian Bottomley[2], Angela Karani[1], Elizabeth Gardiner[1], Rabia Aziza [3], John N. Gitonga[1], Henry Karanja[1], James Nyagwange[1], James Tuju[1], Perpetual Wanjiku[1], Edward Nzomo[4], Evans Kamuri[5], Kaugiria Thuranira[5], Sarah Agunda[5], Gideon Nyutu[1], Anthony O. Etyang[1], Ifedayo M. O. Adetifa[1,2], Eunice Kagucia[1], Sophie Uyoga[1], Mark Otiende[1], Edward Otieno[1], Leonard Ndwiga[1], Charles N. Agoti[1], Rashid A. Aman[6], Mercy Mwangangi[6], Patrick Amoth[6], Kadondi Kasera[6], Amek Nyaguara[1], Wangari Ng'ang'a[7], Lucy B. Ochola[8], Emukule Namdala[9], Oscar Gaunya[9], Rosemary Okuku[9], Edwine Barasa[1,10], Philip Bejon[1,10], Benjamin Tsofa[1], L. Isabella Ochola-Oyier[1], George M. Warimwe[1,10ↄ], Ambrose Agweyu[1ↄ], J. Anthony G. Scott[1,2,10ↄ], Katherine E. Gallagher [1,2ↄ]

1 KEMRI-Wellcome Trust Research Programme, Kilifi, Kenya, 2 Department of Infectious Diseases Epidemiology, London School of Hygiene and Tropical Medicine, London, United Kingdom, 3 School of Life Sciences and the Zeeman Institute for Systems Biology & Infectious Disease Epidemiology Research (SBIDER), University of Warwick, Coventry, United Kingdom, 4 Kilifi County Hospital, Ministry of Health, Government of Kenya, Nairobi, Kenya, 5 Kenyatta National Hospital, Ministry of Health, Government of Kenya, Nairobi, Kenya, 6 Ministry of Health, Government of Kenya, Nairobi, Kenya, 7 Presidential Policy and Strategy Unit, The Presidency, Government of Kenya, Nairobi, Kenya, 8 Institute of Primate Research, Nairobi, Kenya, 9 Busia Country Teaching & Referral Hospital, Busia, Kenya, 10 Nuffield Department of Medicine, Oxford University, Oxford, United Kingdom

ↄ These authors contributed equally to this work.
* RLucinde@kemri-wellcome.org

**Data Availability Statement:** Data are available from the KWTRP Data Governance Committee (contact via dgc@kemri-wellcome.org) for

## Abstract

### Introduction

The high proportion of SARS-CoV-2 infections that have remained undetected presents a challenge to tracking the progress of the pandemic and estimating the extent of population immunity.

### Methods

We used residual blood samples from women attending antenatal care services at three hospitals in Kenya between August 2020 and October 2021and a validated IgG ELISA for SARS-Cov-2 spike protein and adjusted the results for assay sensitivity and specificity. We fitted a two-component mixture model as an alternative to the threshold analysis to estimate of the proportion of individuals with past SARS-CoV-2 infection.

### Results

We estimated seroprevalence in 2,981 women; 706 in Nairobi, 567 in Busia and 1,708 in Kilifi. By October 2021, 13% of participants were vaccinated (at least one dose) in Nairobi,

researchers who meet the criteria for access. Criteria include: The requestor has a disclosed hypothesis and research question that can be answered using the data. The requestor is affiliated with a reputable research organisation, which has capacity to store and analyse the data according to good clinical practice / good data management practice.

**Funding:** This project was funded by the Wellcome Trust (grants 220991/Z/20/Z and 203077/Z/16/Z), the Bill and Melinda Gates Foundation (INV-017547), and the Foreign Commonwealth and Development Office (FCDO) through the East Africa Research Fund (EARF/ITT/039) and is part of an integrated programme of SARS-CoV-2 sero-surveillance in Kenya led by KEMRI Wellcome Trust Research Programme. A.A. is funded by a DFID/MRC/NIHR/Wellcome Trust Joint Global Health Trials Award (MR/R006083/1), J.A.G.S. is funded by a Wellcome Trust Senior Research Fellowship (214320) and the NIHR Health Protection Research Unit in Immunisation, I.M.O.A. is funded by the United Kingdom's Medical Research Council and Department For International Development through an African Research Leader Fellowship (MR/S005293/1) and by the NIHR-MPRU at UCL (grant 2268427 LSHTM). G.M.W. is supported by a fellowship from the Oak Foundation. C.N.A. is funded by the DELTAS Africa Initiative [DEL-15-003], and the Foreign, Commonwealth and Development Office and Wellcome (220985/Z/20/Z). S.U. is funded by DELTAS Africa Initiative [DEL-15-003], L.I.O.-O. is funded by a Wellcome Trust Intermediate Fellowship (107568/Z/15/Z). R.A is funded by National Institute for Health Research (NIHR) (project reference 17/63/82) using UK aid from the UK Government to support global health research. The views expressed in this publication are those of the authors and not necessarily those of the funding agencies. The funders had no role in study design, data collection and analysis, decision to publish, or preparation of the manuscript.

**Competing interests:** This project was funded by a commercial source, the Wellcome Trust (grants 220991/Z/20/Z and 203077/Z/16/Z). This does not alter our adherence to PLOS ONE policies on sharing data and materials. The funders had no role in study design, data collection and analysis, decision to publish, or preparation of the manuscript. Authors declare no other competing interests.

2% in Busia. Adjusted seroprevalence rose in all sites; from 50% (95%CI 42–58) in August 2020, to 85% (95%CI 78–92) in October 2021 in Nairobi; from 31% (95%CI 25–37) in May 2021 to 71% (95%CI 64–77) in October 2021 in Busia; and from 1% (95% CI 0–3) in September 2020 to 63% (95% CI 56–69) in October 2021 in Kilifi. Mixture modelling, suggests adjusted cross-sectional prevalence estimates are underestimates; seroprevalence in October 2021 could be 74% in Busia and 72% in Kilifi.

## Conclusions

There has been substantial, unobserved transmission of SARS-CoV-2 in Nairobi, Busia and Kilifi Counties. Due to the length of time since the beginning of the pandemic, repeated cross-sectional surveys are now difficult to interpret without the use of models to account for antibody waning.

## Introduction

Globally, as countries are confronted with new waves of SARS-CoV-2 infections and new variants, WHO recommendations have focused on enhancing population immunity with the available COVID-19 vaccines [1]. In Kenya, as with many lower-middle income countries, COVID-19 vaccine supplies have been limited [2]. Vaccination started in March 2021 and by 31st October 2021, 3.7 million people had received their first dose (13.5% of the adult population), 1.6 million had received their second dose (6% of the adult population), with some geographic heterogeneity. In Nairobi 34% were partially vaccinated, 18% were fully vaccinated, compared to Busia where 8% were partially vaccinated, 3% fully vaccinated, and Kilifi where 5% were partially vaccinated, 2% fully vaccinated [3].

As vaccine coverage and vaccine-induced immunity is still considered to be low in Kenya, it remains important to track the potential protection conferred by natural infection. By the 31st October 2021, Kenya had experienced four waves of infections, and reported a total of 253,310 confirmed cases and 5281 deaths. However, with just 3.9% of the population over 65 years of age [4], the proportion of infections that have been asymptomatic is likely to be very high [5]. Additionally, limited access to tests and low uptake of testing makes it likely that a substantial proportion of cases have remained undetected. Measuring the prevalence of antibodies to SARS-CoV-2 is an alternative way to estimate the cumulative incidence of infection. A number of serological assays have been developed and perform well with high sensitivity and specificity [6–9]. We have shown that 5.2% of blood donors in Kenya had SARS-CoV-2 antibodies in June 2020 and this had risen to 9.1% in September 2020 and 48.5% by March 2021 [10–12]. However, it is unclear whether blood donors are representative of the population.

In the context of a pandemic, sentinel public health surveillance using residual aliquots of routinely collected blood samples has the potential to overcome participation bias. For example, sero-surveillance for HIV among women attending antenatal care was used to track the progress of the HIV pandemic and showed prevalence estimates that were similar to population samples from the same areas [13,14]. It remains unclear whether pregnancy alters susceptibility to SARS-CoV-2 infection [15]; however, a large systematic review has found no difference in risk of becoming symptomatic when comparing pregnant women with confirmed SARS-CoV-2 infection in women of the same age [16,17].

In Kenya, in 2014, 50% of women had had at least one pregnancy or were pregnant by 20 years of age and the coverage of at least one antenatal care visit was 96% [18]. Residual blood samples from mothers visiting antenatal care for the first time may therefore represent a relatively unbiased sample of young women, and an alternative sentinel surveillance population to blood donors. Testing an aliquot of blood for antibodies to SARS-CoV-2 is feasible as a venous blood sample (5ml) is already taken to screen mothers for malaria, HIV and syphilis at their first antenatal visit. We aimed to determine the prevalence of antibodies against SARS-CoV-2 in mothers attending antenatal care at three referral hospitals in Kenya.

## Methods

### Setting

In a collaboration between the Kenyan Ministry of Health (MOH) and KEMRI-Wellcome Trust Research Programme (KWTRP), three referral hospitals were engaged. Kenyatta National Hospital (KNH) is the national referral tertiary hospital located in Nairobi, the country's capital city, approximately 3km from the central business district. The population of Nairobi city was 4,397,073 in 2019 [4]. Busia Country Teaching & Referral Hospital (BCTRH) serves Busia County, an area of 1628 $km^2$, with a population of 893,681 (548/ $km^2$). Kilifi County Hospital (KCH) is the county referral hospital in Kilifi Town. Kilifi county covers an area of 12,000 $km^2$, with a predominantly rural population of 1.4 million (116/$km^2$) [4].

### Study population

All women attending antenatal care for the first time, who provided a routine blood sample at their clinic visit, were included in the study. Women who did not provide a sample at their first antenatal care visit, or women attending their second or subsequent visit, were excluded.

### Sample collection and processing

In Kenya, a 5ml blood sample is routinely collected at the first antenatal visit. After testing for malaria, syphilis and HIV in the hospital laboratory, the residual volume is usually discarded. In this study, all residual samples were set aside and collected daily for SARS-CoV-2 sero-surveillance. Where possible, the following data were collected from hospital records and linked to the residual sample identity number: date of sample, age, sub-county of residence, trimester of pregnancy, presence or absence of COVID19-like symptoms in the last month and COVID-19 vaccination status ascertained via verbal report confirmed via SMS or certificate. No personal identifiers were collected. All samples were tested at the KWTRP laboratories for IgG to SARS-CoV-2 whole spike protein using an adaptation of the Krammer Enzyme Linked Immunosorbent Assay (ELISA) [6]. Validation of this assay is described in detail elsewhere [10]. Results were expressed as the ratio of test OD to the OD of the plate negative control; samples with OD ratios greater than two were considered positive for SARS-CoV-2 IgG. Sensitivity, estimated in 174 PCR positive Kenyan adults and a panel of 5 sera from the National Institute of Biological Standards in the UK was 92.7% (95% CI 87.9–96.1%); specificity, estimated in 910 serum samples from Kilifi drawn in 2018 was 99.0% (95% CI 98.1–99.5%) [10].

### Analysis

We estimated the proportion of samples seropositive for IgG to SARS-CoV-2. Sampling at least 135 women per month from each hospital would provide estimates of seroprevalence in the range 3–25% with a precision of 3–7%. Bayesian modelling was used to adjust seroprevalence estimates for the sensitivity and specificity of the assay. Non-informative priors were

used for each parameter (sensitivity, specificity and proportion true positive) and the models were fitted using the RStan software package [19] (see S1 Appendix). Sub-county population densities were extracted from the Kenya National Bureau of Statistics' database [4].

To account for the effects of waning IgG in repeated cross-sectional samples, we fitted a two-component mixture model to the $log_2$ OD ratios in unvaccinated individuals. In this model, we assumed that antibody levels follow a normal distribution in previously uninfected individuals and a skew-normal in previously infected individuals. To fit the model, we fixed the standard deviation of the negative component at the value observed in pre-COVID 19 samples. The remaining parameters were estimated using RStan. Details of the priors used in the estimation have been described elsewhere [20].

### Patient and public involvement

The study was conducted as anonymous public health surveillance at the request of the Kenyan MOH, in response to the COVID-19 pandemic. The study directly addressed the needs of the MOH by providing some information on the extent of the spread of SARS-CoV-2 pandemic within Kenya. The public were not involved in the conceptualisation or implementation of this study. The need for individual informed consent from the women whose samples were studied was waived, the protocol was approved by the Scientific and Ethics Review Unit (SERU) of the Kenya Medical Research Institute (Protocol SSC 4085), the Kenyatta National Hospital–University of Nairobi Ethics Review Committee (Protocol P327/06/2020) and the Busia & Kilifi County health management teams.

## Results

### Seroprevalence across time and location

In Nairobi, samples were collected in three rounds: round 1, median date 11[th] August 2020, round 2, median date 22 February 2021, and round 3, median date 30 September 2021. In these time periods, 706 women (93%) provided a sample (S1 Fig). Women were aged between 17 and 45 years (mean 31 years); 275 (40%) attended their first antenatal care visit during their third trimester of pregnancy, although this differed significantly between the rounds: 62% in August 2020, 28% in February 2021 and 35% in September 2021 (p<0.001; S1 Table). A total of 632 (90%) reported residence in 16 different sub-counties of Nairobi, 267 (42%) of mothers were resident in Embakasi North, East or West sub-counties, and 110 (17%) were resident in Dagoretti North or South sub-counties. The proportion of participants living in high vs. low population density sub-counties did not differ by round (S1 Table). Among women who had data on symptoms during the preceding month, 7% reported symptoms in the first two rounds, this significantly differed from the third round where 43% reported symptoms, coinciding with the end of the cold season (June-September). Symptoms were not associated with seropositivity, controlling for age. In Nairobi, seroprevalence, adjusted for the sensitivity and specificity of the ELISA, was 50% in August 2020, 32% in February 2021 and 85% in September 2021 (Table 1). In October 2021, 12.7% of women were vaccinated with at least one dose of COVID-19 vaccine, seroprevalence among the unvaccinated was 82%.

In Busia, samples were collected in 2 rounds: round 1, median date 3[rd] May, and round 2, median date 5[th] October. In this time period a total of 567 first antenatal visits were conducted; 567 (100%) provided a sample (S1 Fig). Women were aged between 14 and 44 (mean age 27 years). Most women (66%) attended their first antenatal care visit in their second trimester, although this differed by round (73% in May and 60% in October (p = 0.007; S1 Table). In May, 40% of women reported symptoms in the last month, which differed significantly from October 2021 where 56% reported symptoms. Symptoms were not associated with

**Table 1. Seroprevalence of IgG to SARS-CoV-2 among mothers attending antenatal care in Kenyatta National Hospital (KNH), Nairobi.**

| KNH | 30th July– 25th August 2020 | | | | 27th Jan- 11th March 2021 | | | | 7th September-19th October 2021 | | | |
|---|---|---|---|---|---|---|---|---|---|---|---|---|
| | Seroprevalence | | Adjusted seroprevalence | | Seroprevalence | | Adjusted seroprevalence | | Seroprevalence | | Adjusted seroprevalence | |
| Nairobi | n / N | % | % | 95% CI | n / N | % | % | 95% CI | n / N | % | % | 95% CI |
| All | 91 / 196 | 46.4 | 49.9 | 42.1–58.2 | 80 / 265 | 30.2 | 32.1 | 26.2–38.4 | 193 / 245 | 78.8 | 84.9 | 78.3–91.5 |
| Age | | | | | | | | | | | | |
| First 17–29 years | 39 / 93 | 41.9 | 44.9 | 33.8–56.9 | 28 / 101 | 27.7 | 29.6 | 20.7–39.2 | 82 / 101 | 81.2 | 87.2 | 77.9–95.7 |
| 30–45 years | 44 / 90 | 48.9 | 52.5 | 41.1–63.7 | 47 / 141 | 33.3 | 35.7 | 27.3–44.3 | 107 / 139 | 77.0 | 83.0 | 74.0–91.7 |
| Trimester | | | | | | | | | | | | |
| First | 7 / 17 | 41.2 | 44.9 | 21.2–70.1 | 27 / 83 | 32.5 | 35.0 | 24.1–46.4 | 48 / 60 | 80.0 | 85.5 | 73.0–96.1 |
| Second | 21 / 53 | 39.6 | 42.9 | 29.8–57.4 | 34 / 106 | 32.1 | 34.5 | 25.1–44.3 | 77 / 96 | 80.2 | 86.1 | 76.0–95.5 |
| Third | 58 / 114 | 50.9 | 54.7 | 44.6–64.6 | 18 / 75 | 24.0 | 25.8 | 16.4–37.0 | 64 / 84 | 76.2 | 81.6 | 71.2–91.7 |
| Any symptoms in last month* | | | | | | | | | | | | |
| Yes | 7 / 12 | 58.3 | 61.2 | 33.3–86.1 | 3 / 18 | 16.7 | 20.8 | 5.5–42.2 | 80 / 106 | 75.5 | 81.2 | 71.3–90.5 |
| No | 78 / 172 | 45.3 | 48.7 | 40.2–57.5 | 77 / 247 | 31.2 | 33.3 | 26.9–40.0 | 113 / 139 | 81.3 | 87.5 | 79.3–95.1 |
| Population density of sub-county of residence | | | | | | | | | | | | |
| <20000/km² | 44 / 97 | 45.4 | 48.8 | 38.1–59.4 | 29 / 104 | 27.9 | 29.8 | 20.6–39.8 | 86 / 102 | 84.3 | 90.6 | 82.2–97.9 |
| 20-81000/km² | 39 / 79 | 49.4 | 53.1 | 41.0–65.7 | 40 / 124 | 32.3 | 34.6 | 25.9–44.1 | 85 / 115 | 73.9 | 79.5 | 70.3–88.5 |
| COVID-19 vaccine status† | | | | | | | | | | | | |
| Vaccinated | - | - | - | - | - | - | - | - | 30 / 31 | 96.8 | 96.1 | 86.2–99.9 |
| Unvaccinated | - | - | - | - | - | - | - | - | 163 / 214 | 76.2 | 82.1 | 75.1–89.2 |

* Women were asked about the full list of COVID-19 symptoms as per the MOH COVID-19 screening form i.e. fever/ chills, general weakness, cough, sore throat, runny nose, shortness of breath, diarrhoea, nausea/ vomiting, headache, irritability/ confusion, pain (muscular/ chest/ abdominal/ joint).

† Vaccination status (at least one dose) was not available for the first two rounds of data collection, vaccination began in Kenya in March 2021 and at first targeted specific groups only, we assume that the vaccine coverage among women attended antenatal care between 27th Jan-11th March was 0%.

Variations in seroprevalence by any of the explanatory variables were not statistically significant in any time period when tested with chi2 test.

seropositivity, controlling for age. Adjusted seroprevalence in Busia increased from 31% in May 2021 to 71% in October 2021 (Table 2). Just 6 (2%) of women were vaccinated in October 2021, and seroprevalence remained 71% among the unvaccinated.

In Kilifi, 1707 samples were collected between the 18th September 2020 and 22nd October 2021, collection was continuous apart from during the healthcare worker strike (December 2020 to February 2021). No data were available on age, trimester, location, symptoms or COVID-19 vaccination status. Adjusted seroprevalence increased over the period of sample collection from 1% in September 2020 to 63% in October 2021 (p = 0.0001, Chi sq test for trend; Table 3).

## Mixture model results

When two distinct distributions were fitted to the data—corresponding to antibody levels in previously infected and previously uninfected individuals—there were substantial overlaps in the distributions, especially at low seroprevalences (Fig 1). As seroprevalence estimates increased, the distributions became more distinct. The mixture model produced estimates of cumulative incidence that were consistently higher than those of the threshold analysis except for the final round in Nairobi, where the results of both analyses were the same (85%) and the distributions hardly overlapped (85%; Fig 2). Median OD ratios among the unvaccinated, seropositive individuals increase over time in all three areas, potentially indicating natural boosting through re-infections (Fig 1, S2 Table).

**Table 2. Seroprevalence of IgG to SARS-CoV-2 among mothers attending antenatal care in Busia County Teaching & Referral Hospital (BCTRH), Busia.**

| BCTRH, Busia | | 15th April—21 May 2021 | | | | 20 September– 22 October 2021 | | | |
|---|---|---|---|---|---|---|---|---|---|
| | | Seroprevalence | | Adjusted seroprevalence | | Seroprevalence | | Adjusted seroprevalence | |
| | | n / N | % | % | 95% CI | n / N | % | % | 95% CI |
| All | | 78 / 270 | 28.9 | 30.6 | 24.7–37.1 | 195 / 297 | 65.7 | 70.7 | 64.1–77.4 |
| Age | | | | | | | | | |
| | 17–29 years | 3 / 14 | 21.4 | 26.0 | 6.5–51.4 | 128 / 203 | 63.1 | 67.8 | 59.8–76.0 |
| | 30–45 years | 2 / 7 | 28.6 | 35.3 | 7.4–71.3 | 58 / 84 | 69.0 | 74.2 | 62.7–85.4 |
| Trimester | | | | | | | | | |
| | First | 18 / 51 | 35.3 | 38.2 | 25.1–52.6 | 50 / 78 | 64.1 | 68.7 | 56.6–80.1 |
| | Second | 50 / 192 | 26.0 | 27.7 | 21.3–34.7 | 114 / 175 | 65.1 | 70.1 | 61.8–78.4 |
| | Third | 7 / 22 | 31.8 | 35.3 | 17.3–55.8 | 26 / 39 | 66.7 | 71.2 | 54.3–86.4 |
| Any symptoms in last month* | | | | | | | | | |
| | Yes | 37 / 109 | 33.9 | 36.4 | 26.6–46.2 | 107 / 164 | 65.2 | 70.1 | 61.6–78.8 |
| | No | 41 / 161 | 25.5 | 27.0 | 19.6–35.1 | 84 / 129 | 65.1 | 70.0 | 60.4–79.3 |
| COVID-19 vaccination status† | | | | | | | | | |
| | Vaccinated | - | - | - | - | 4 / 6 | 66.7 | 67.0 | 29.8–96.1 |
| | Unvaccinated | - | - | - | - | 188 / 288 | 65.3 | 70.5 | 63.8–77.3 |

[1] DOB was only available for 22 women in the first round of data collection.

* Women were asked about the full list of COVID-19 symptoms as per the MOH COVID-19 screening form i.e. fever/ chills, general weakness, cough, sore throat, runny nose, shortness of breath, diarrhoea, nausea/ vomiting, headache, irritability/ confusion, pain (muscular/ chest/ abdominal/ joint).

† Vaccination status (at least one dose) was not available for the first two rounds of data collection, vaccination began in Kenya in March 2021 and at first targeted specific groups only, we assume that the vaccine coverage among women attended antenatal care between 27th Jan-11th March was 0%.

Variations in seroprevalence by any of the explanatory variables were not statistically significant in any time period when tested with chi2 test.

## Discussion

Surveillance for IgG antibodies to SARS-CoV-2 among mothers attending antenatal services in three county referral hospitals in Kenya has revealed evidence of a substantial amount of prior infection by October 2021. Seroprevalence is currently highest in Nairobi, then Busia, then Kilifi, correlating with the counties' population densities.

**Table 3. Seroprevalence of IgG to SARS-CoV-2 among mothers attending antenatal care in Kilifi County Hospital (KCH), over time.**

| KCH, Kilifi[1] | | | Seroprevalence | | Adjusted seroprevalence | |
|---|---|---|---|---|---|---|
| | | | n / N | % | % | 95% CI |
| Month | Sept-Oct 2020 | | 3 / 265 | 1.1 | 0.9 | 0.0–2.7 |
| | Nov-Dec 2020 | | 32 / 236 | 13.6 | 14.0 | 9.4–19.5 |
| | Mar-Apr 2021 | | 55 / 260 | 21.2 | 22.2 | 16.7–28.1 |
| | May-Jun 2021 | | 104 / 382 | 27.2 | 28.9 | 23.9–34.4 |
| | Jul-Aug 2021 | | 148 / 260 | 56.9 | 61.2 | 54.4–68.4 |
| | Sept-Oct 2021 | | 178 / 305 | 58.4 | 62.7 | 56.2–69.1 |

[1] No age, trimester or symptom data were available from the antenatal care records at KCH. Months were combined into 2-month batches due to low numbers.

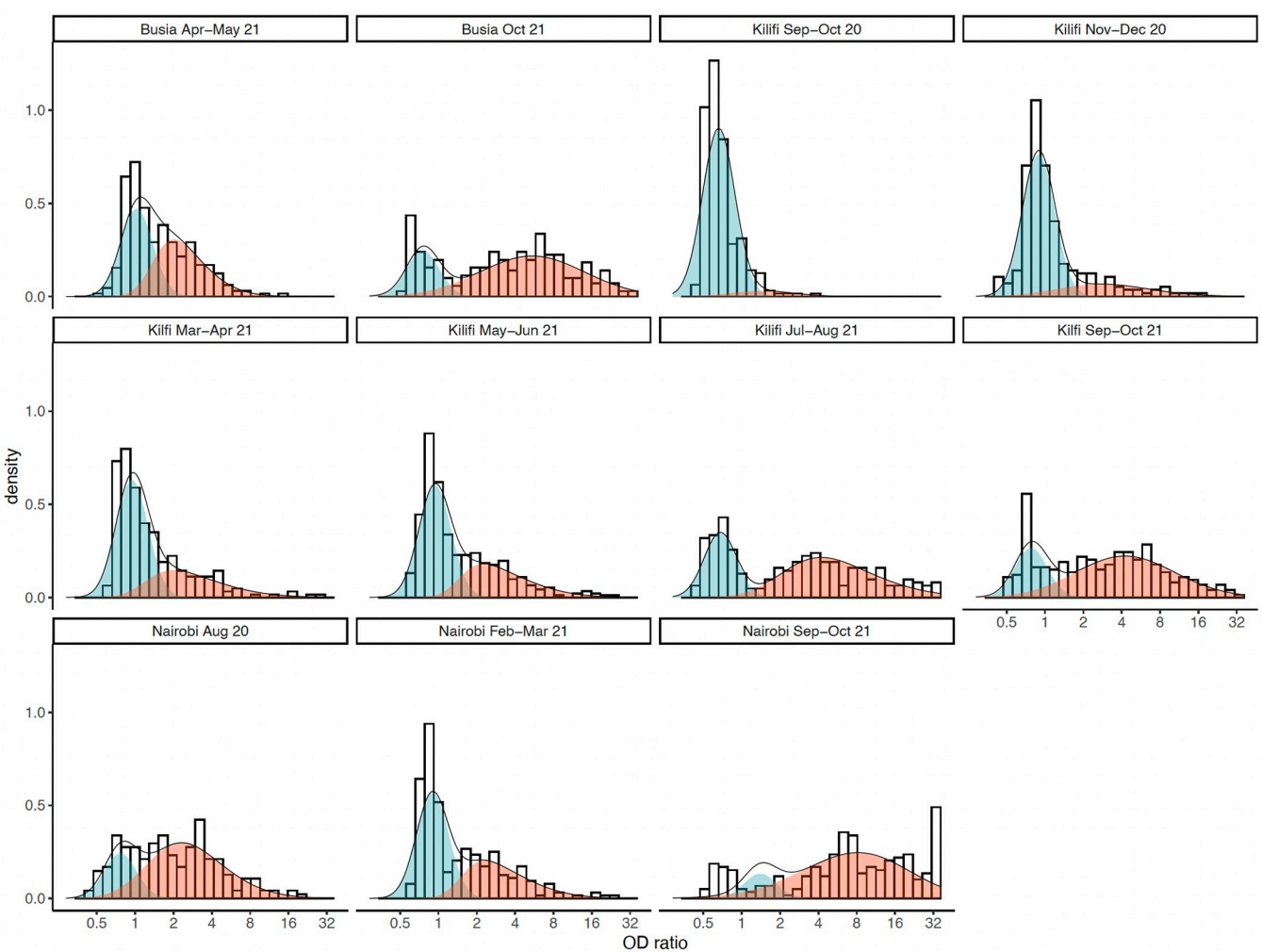

**Fig 1. Mixture distributions fitted to anti-spike IgG antibody data collected in Kilifi (KCH), Busia and Nairobi (KNH).** The red distribution represents predicted responses in individuals previously infected with SARS-CoV-2 and the blue distribution represents predicted responses in previously uninfected individuals.

In Nairobi, in August 2020, just after the peak of the first wave of SARS-CoV-2 infections, mixture modelling, which attempts to account for the wide range of OD ratios among those exposed to the virus better than the simple threshold analysis [20], indicates a cumulative incidence of 75%, just 4 months after the start of the pandemic. At the same timepoint, 6,727 PCR-confirmed infections had been registered across the city (<1% of the County's population; S2 Fig). In March 2021, a year after the pandemic began, seroprevalence was lower, 32%. This second group of women reported residing in the same sub-counties and were on average the same age (S1 Table). The high levels of transmission of the virus in these locations early in the pandemic may have meant some of these women had been infected at some point in the last year, but had since seroreverted. Data on the rate of seroreversion differs with the assay used [21] and the severity of the initial infection [22]; approximately 9–12% with mild symptoms may sero-revert 4–6 months post-infection [23,24]. Additionally, such high seroprevalence earlier in the year may have reduced the number susceptible and dampened transmission within the same communities by March 2021. Modelling indicates the first wave could have predominantly affected communities of low-income earners, more likely to use

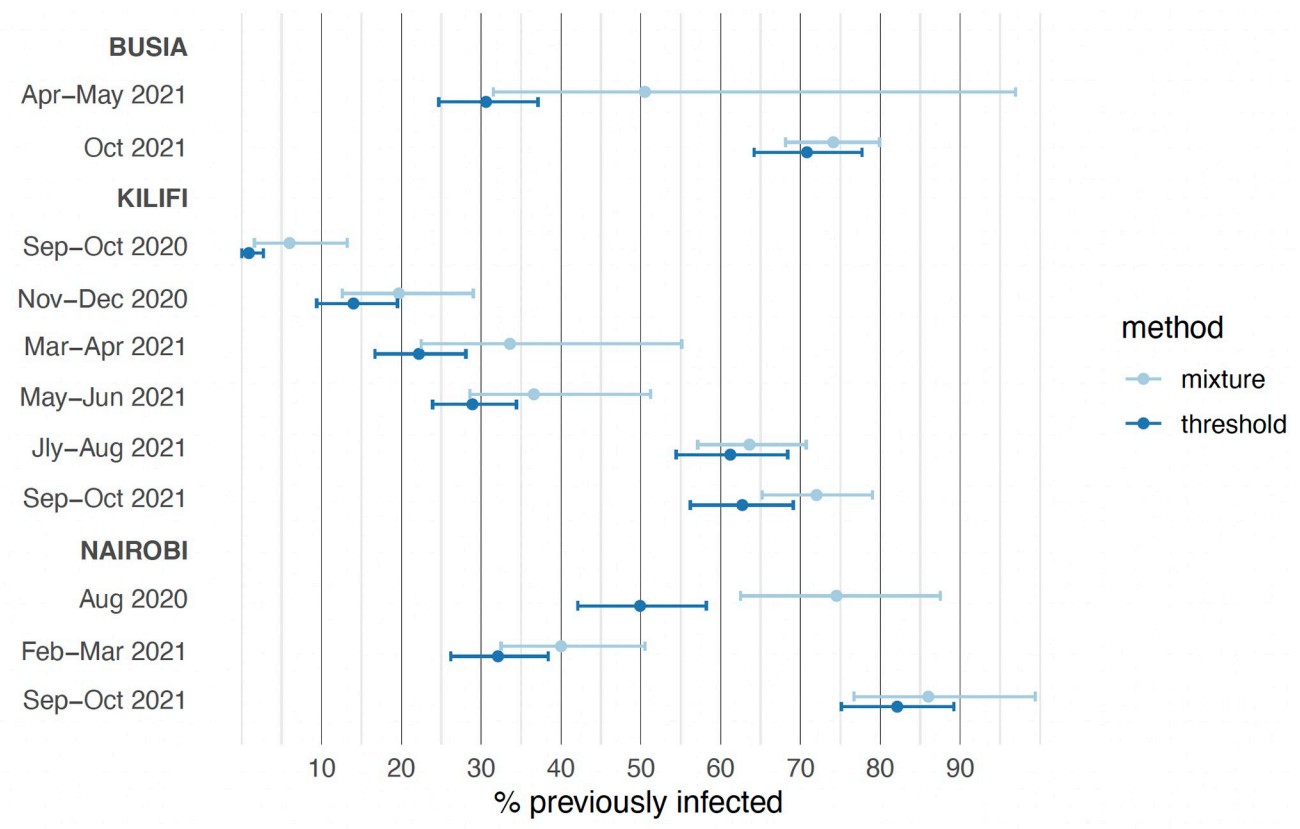

**Fig 2. Adjusted and modelled estimates of the cumulative incidence of SARS-CoV-2 infection.** Estimates are shown with 95% credible intervals.

public hospitals, such as KNH, with the later waves affecting a group including higher-income, private healthcare users [25]. In October 2021, seroprevalence in a third group of women was 85%, 76% among those unvaccinated. These women were very similar in age, residence location and trimester to the second group. The survey was conducted just after the fourth wave of cases in Kenya, natural boosting could have occurred in these communities if residents encountered the virus repeatedly [26]. This is supported by the higher median OD ratios among unvaccinated, seropositive individuals in the third round compared to both of the previous rounds. The proportion vaccinated with at least one dose (13%) is lower than the average for Nairobi adults at this time of 34% [3], although nationally only 13.5% of the population were vaccinated with one dose at this time point.

In Busia, seroprevalence in May 2021 was 31% using the threshold analysis, and 51% using the mixture model analysis. The two distributions of OD ratios had a substantial amount of overlap leading to uncertainty in the estimate from mixture modelling at this time point. Threshold seroprevalence increased to 71% in October 2021. By 21st October 2021 only 2% of our study population were vaccinated with at least one dose, lower than the nationally reported coverage of 8% in Busia [3]. Western Kenya was affected in the fourth wave of the pandemic in July-September 2021 and the seroprevalence represents a substantial amount of natural infection.

Seroprevalence steadily increased in Kilifi during the sampling timeframe to 63% (using the threshold analysis) or 72% (using the mixture model analysis) in October 2021. The slower increase in seroprevalence in Kilifi compared to Nairobi is consistent with modelling

suggesting that the initial wave of the COVID-19 pandemic was concentrated in urban centres, with subsequent spread increasingly affecting rural areas [25]; Kilifi County reported a marked increase in the number of infections in December 2020 (S2 Fig).

A strength of this analysis is the use of a rigorously validated serological assay, using locally relevant control populations and reference panels from the National Institute for Biological Standards and Control (NIBSC) in the UK [27]. The threshold used to define seropositivity was chosen to prioritise specificity over sensitivity, i.e. to minimize the number of false positives. The very low crude seroprevalence (0/82; 0%) in the first month of samples from Kilifi adds confirmation that the specificity of this assay is very high.

Although sero-surveillance among pregnant women has been used as a proxy for population-based SARS-CoV-2 surveillance in high income countries [28–34], the representativeness of the sample in Kenya is unknown. A national survey in 2014 indicated 18% of the population utilised public hospitals at their last visit to outpatient services (a further 40% utilized public health centres or dispensaries). Utilisation of public health services was correlated with lower education levels [35].

The seroprevalence estimates from women attending antenatal care differ from the seroprevalence estimates available from blood donor samples. The Nairobi seroprevalence of 50% in August is substantially higher than the 10% seroprevalence reported among blood donors in the same county in June-August 2020 [10], however in March 2021, 32% seroprevalence at antenatal care was lower than the estimated 62% seroprevalence in blood donors [12]. The majority of expectant mothers attending care in KNH consistently came from 5 sub-counties close to the hospital, which are densely populated with low-income earners [4]. Blood transfusion donors are likely to be more heterogenous and widely distributed across Nairobi including areas of lower population density and greater affluence. The estimates from antenatal care in Kilifi in September 2020 (1%) and April 2021 (22%) were lower than the 14.1% seroprevalence reported among blood donors from Coastal Counties in September 2020 and the 43% seroprevalence among blood donors in Jan-March 2021 [10–12]. Blood donations come from across the county including urban centres such as Malindi, whereas women attending antenatal care in Kilifi represent a less heterogenous semi-urban group. It is clear that viral transmission has been heterogenous in terms of geography and socioeconomic status, seroprevalence estimates from multiple different sentinel populations provide more reliable indicators of the development of the pandemic than any one estimate alone.

The impact of pregnancy on susceptibility to SARS-CoV-2 infection is unclear [15]; however, comparisons of infected pregnant women with non-pregnant women of the same age suggests that a similar proportion of infections become symptomatic [16]. This would suggest a similar proportion of pregnant and non-pregnant women mount a protective antibody response and seroprevalence estimates are generalisable to non-pregnant women of the same age. In a comparison of samples from blood donors and antenatal women in Australia, the two sample sets estimated seroprevalence within 0.1% of each other, although overall prevalence was very low [36]. Additionally, seroprevalence in blood donors in Kenya did not differ by sex [10], suggesting that these results from pregnant women may be generalisable to men between 17–45 years of age, residing in the same areas.

Our analysis is constrained by the nature of the anonymised surveillance data available. Data on age, trimester and location for the women in Kilifi would have allowed more valid comparisons with data from other sources. It is difficult to assess how comparable the different rounds from the same location are, without more data. As discussed, we lack local data on the rate of antibody waning, which is important to estimate cumulative incidence of infection from snapshot seroprevalence estimates [37,38]. This is especially important in populations, like those reported here, where ongoing transmission may cause 'natural boosting' [37,39,40].

## Conclusions

This seroprevalence study of women attending antenatal clinics suggests there has been substantial, unobserved transmission of SARS-CoV-2 within communities in Nairobi, Busia and Kilifi Counties. However, it is becoming difficult to interpret the results of cross-sectional seroprevalence studies due to the length of the pandemic [41]. To attempt to account, to some extent, for antibody waning, we have used mixture modelling, this suggests that 85% of the population using a public hospital in Nairobi have been previously infected with SARS-COV-2. At least in the short-term, asymptomatic infection is protective [26] and these seroprevalence estimates should be taken into account when estimating population level immunity. Increases in antibody concentration over time implies an increasing level of population protection that may be attributable to reinfections.

## Supporting information

**S1 Fig. Ante-natal care sample flow.**
(DOCX)

**S2 Fig.** A) Daily (faint lines) and the 7-day moving average (bold lines) number of positive PCR tests per million population and B) Cumulative daily number of PCR positive tests for SARS-CoV-2 in Kenya, per million population.
(DOCX)

**S1 Table. Comparability of the sample sets, by round.** (A) Kenyatta National Hospital data (B) Busia Country Teaching & Referral Hospital.
(DOCX)

**S2 Table. Median OD ratios among unvaccinated, seropositive participants over time.**
(DOCX)

**S1 Appendix. Stan code and input data for estimating adjusted seroprevalence.**
(DOCX)

## Acknowledgments

We thank the Kenyatta National Hospital, Busia Country Teaching & Referral Hospital and Kilifi County Hospital employees who collected the samples during routine antenatal care visits and the women themselves for providing samples for routine health screening. We thank Rebeccah Ayako, Evalyne Akinyi and Cedrick Shikoli at the Institute of Primate Research for processing the samples from KNH. We thank F. Krammer for providing the plasmids used to generate the spike protein used in this work. Development of SARS-CoV-2 reagents was partially supported by the NIAID Centres of Excellence for Influenza Research and Surveillance (CEIRS) contract HHSN272201400008C. The COVID-19 convalescent plasma panel (NIBSC 20/118) and research reagent for SARS-CoV-2 Ab (NIBSC 20/130) were obtained from the NIBSC, UK. We also thank the WHO SOLIDARITY II network for sharing of protocols and for facilitating the development and distribution of control reagents. This paper has been published with the permission of the director, Kenya Medical Research Institute.

## Author Contributions

**Conceptualization:** Anthony O. Etyang, Ifedayo M. O. Adetifa, Eunice Kagucia, Sophie Uyoga, George M. Warimwe, Ambrose Agweyu, J. Anthony G. Scott, Katherine E. Gallagher.

**Data curation:** Gideon Nyutu.

**Formal analysis:** Christian Bottomley, Katherine E. Gallagher.

**Funding acquisition:** George M. Warimwe, Ambrose Agweyu, J. Anthony G. Scott.

**Investigation:** Ruth K. Lucinde, Daisy Mugo, Angela Karani, Elizabeth Gardiner, John N. Gitonga, Henry Karanja, James Nyagwange, James Tuju, Perpetual Wanjiku, Edward Nzomo, Evans Kamuri, Kaugiria Thuranira, Sarah Agunda, Edward Otieno, Leonard Ndwiga, Charles N. Agoti, Amek Nyaguara, Lucy B. Ochola, Emukule Namdala, Oscar Gaunya, Rosemary Okuku, L. Isabella Ochola-Oyier.

**Methodology:** Christian Bottomley, Katherine E. Gallagher.

**Project administration:** Ruth K. Lucinde.

**Supervision:** L. Isabella Ochola-Oyier, George M. Warimwe, Ambrose Agweyu, J. Anthony G. Scott, Katherine E. Gallagher.

**Validation:** George M. Warimwe.

**Visualization:** Christian Bottomley, Rabia Aziza.

**Writing – original draft:** Ruth K. Lucinde, Katherine E. Gallagher.

**Writing – review & editing:** Ruth K. Lucinde, Daisy Mugo, Christian Bottomley, Angela Karani, Elizabeth Gardiner, Rabia Aziza, John N. Gitonga, Henry Karanja, James Nyagwange, James Tuju, Perpetual Wanjiku, Edward Nzomo, Evans Kamuri, Kaugiria Thuranira, Sarah Agunda, Gideon Nyutu, Anthony O. Etyang, Ifedayo M. O. Adetifa, Eunice Kagucia, Sophie Uyoga, Mark Otiende, Edward Otieno, Leonard Ndwiga, Charles N. Agoti, Rashid A. Aman, Mercy Mwangangi, Patrick Amoth, Kadondi Kasera, Amek Nyaguara, Wangari Ng'ang'a, Lucy B. Ochola, Emukule Namdala, Oscar Gaunya, Rosemary Okuku, Edwine Barasa, Philip Bejon, Benjamin Tsofa, L. Isabella Ochola-Oyier, George M. Warimwe, Ambrose Agweyu, J. Anthony G. Scott, Katherine E. Gallagher.

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
