## [Decision Letter · Decision Letter 0]

27 May 2022

PONE-D-22-06158Sero-surveillance for IgG to SARS-CoV-2 at antenatal care clinics in three Kenyan referral hospitals: repeated cross-sectional surveys 2020-21PLOS ONE

Dear Dr. Gallagher,

Thank you for submitting your manuscript to PLOS ONE. After careful consideration, we feel that it has merit but does not fully meet PLOS ONE’s publication criteria as it currently stands. Therefore, we invite you to submit a revised version of the manuscript that addresses the points raised during the review process.

We look forward to receiving your revised manuscript.

Kind regards,

Francesco Maria Galassi, MD MRSB MCSFS FRSPH

Academic Editor

PLOS ONE

Journal Requirements:

“This project was funded by the Wellcome Trust (grants 220991/Z/20/Z and 203077/Z/16/Z), the Bill and Melinda Gates Foundation (INV-017547), and the Foreign Commonwealth and Development Office (FCDO) through the East Africa Research Fund (EARF/ITT/039) and is part of an integrated programme of SARS-CoV-2 sero-surveillance in Kenya led by KEMRI Wellcome Trust Research Programme.

A.A. is funded by a DFID/MRC/NIHR/Wellcome Trust Joint Global Health Trials Award (MR/R006083/1), J.A.G.S. is funded by a Wellcome Trust Senior Research Fellowship (214320) and the NIHR Health Protection Research Unit in Immunisation, I.M.O.A. is funded by the United Kingdom’s Medical Research Council and Department For International Development through an African Research Leader Fellowship (MR/S005293/1) and by the NIHR-MPRU at UCL (grant 2268427 LSHTM). G.M.W. is supported by a fellowship from the Oak Foundation. C.N.A. is funded by the DELTAS Africa Initiative [DEL-15-003], and the Foreign, Commonwealth and Development Office and Wellcome (220985/Z/20/Z). S.U. is funded by DELTAS Africa Initiative [DEL-15-003], L.I.O.-O. is funded by a Wellcome Trust Intermediate Fellowship (107568/Z/15/Z). R.A is funded by National Institute for Health Research (NIHR) (project reference 17/63/82) using UK aid from the UK Government to support global health research.

The views expressed in this publication are those of the authors and not necessarily those of the funding agencies”

“This project was funded by the Wellcome Trust (grants 220991/Z/20/Z and 203077/Z/16/Z), the Bill and Melinda Gates Foundation (INV-017547), and the Foreign Commonwealth and Development Office (FCDO) through the East Africa Research Fund (EARF/ITT/039) and is part of an integrated programme of SARS-CoV-2 sero-surveillance in Kenya led by KEMRI Wellcome Trust Research Programme. The funders had no role in study design, data collection and analysis, decision to publish, or preparation of the manuscript.”

6. Thank you for stating the following in the Competing Interests/Financial Disclosure * (delete as necessary) section:

“This project was funded by the Wellcome Trust (grants 220991/Z/20/Z and 203077/Z/16/Z), the Bill and Melinda Gates Foundation (INV-017547), and the Foreign Commonwealth and Development Office (FCDO) through the East Africa Research Fund (EARF/ITT/039) and is part of an integrated programme of SARS-CoV-2 sero-surveillance in Kenya led by KEMRI Wellcome Trust Research Programme. The funders had no role in study design, data collection and analysis, decision to publish, or preparation of the manuscript.”

We note that you received funding from a commercial source: “Wellcome Trust”

Reviewers' comments:

Reviewer's Responses to Questions

**Comments to the Author**

1. Is the manuscript technically sound, and do the data support the conclusions?

Reviewer #1: Yes

Reviewer #2: Yes

2. Has the statistical analysis been performed appropriately and rigorously? 

Reviewer #1: Yes

Reviewer #2: Yes

3. Have the authors made all data underlying the findings in their manuscript fully available?

Reviewer #1: Yes

Reviewer #2: Yes

4. Is the manuscript presented in an intelligible fashion and written in standard English?

Reviewer #1: Yes

Reviewer #2: Yes

5. Review Comments to the Author

Reviewer #1: This is a succint report of seropositivity to SARS-CoV-2 in Kenya. Well executed and well presented. My major comment is that the infection with the virus, as evidenced by the presence of antibodies in blood samples, is considered the prevalence of the disease. Many patients were asymptomatic, in some the antibodies could be effective at fighting infection and thus preventing any symptoms. This comment is not directed specifically at the Authors, but refers to the general confusion of the disease with the infection by its causative agent. Infection does not mean disease (presence of disrupted organismal homeostasis).

In the "Discussion" the Authors repeat details of the results unnecessarily. The last sentence of the Discussion (lines 319-320) is insightful.

As a South African who lived through the years of the downfall of the apartheid, I found the use of the acronym ANC confusing (African National Congress) and the other acronyms interferring with understanding of the text. Thre is no list of abbreviations, but I would encourage Authors to avoid the use of acronyms in a publication on-line whose cost is not affected by the number of words, or characters, used in the text.

Reviewer #2: The authors of this manuscript, entitled “Sero-surveillance for IgG to SARS-CoV-2 at antenatal care clinics in three Kenyan referral hospitals: repeated cross-sectional surveys 2020-21”, presented a cross-sectional study aimed at assessing the transmission of SARS-CoV-2 in Kenya. The manuscript fits the aim of “PLOS ONE,” which publishes papers and research in the area of the natural sciences, medical research, engineering, and the related social sciences and humanities.

The title, as well as the abstract, are informative; nevertheless, the study’s aim has not been clearly described and needs to be elucidated in more detail,

Introduction requires a deeper contextualization of SARS-CoV-2 infection

Finally, the text should be written in a scientific style containing passive voice and 3rd person.

Moreover, check the references list and the references cited in the text. Please review them for completeness and errors and note that according to PLOS ONE, the references must be cited in the Vancouver style.

Therefore, I strongly recommend accepting the manuscript after minor revision.

6. PLOS authors have the option to publish the peer review history of their article (what does this mean?). If published, this will include your full peer review and any attached files.

Reviewer #1: **Yes: **Maciej Henneberg

Reviewer #2: No

---

## [Author Response · Author response to Decision Letter 0]

21 Jul 2022

Editorial comments: 

Response: We apologise for this oversight. We have gone through the document and amended the formatting of the title page, headings, figures and tables accordingly. 

2. We note that you have included the phrase “data not shown” in your manuscript. Unfortunately, this does not meet our data sharing requirements. 

Response: We apologise for the confusion - the data the statement refers to is included in the full dataset which is available in the repository as indicated, we have removed ‘data not shown’. 

3. Please include captions for your Supporting Information files at the end of your manuscript, and update any in-text citations to match accordingly.

Response: Done 

4. In your Data Availability statement (online form), you have not specified where the minimal data set underlying the results described in your manuscript can be found. Upon re-submitting your revised manuscript, please upload your study’s minimal underlying data set as either Supporting Information files or to a stable, public repository and include the relevant URLs, DOIs, or accession numbers within your revised cover letter

Response: Data is deposited in the institutional data repository as per KWTRP policies. Please include the following statement in the manuscript: 

“ Data are available upon request from the KWTRP online data repository by submitting a data request form to the KWTRP Data Governance committee here: https://dataverse.harvard.edu/dataverse/kwtrp”

5. Please remove any funding-related text from the manuscript and let us know how you would like to update your Funding Statement

Response: We have removed the funding statement from the manuscript. Please amend the funding statement in the application portal to the statement that was in the manuscript: 

“This project was funded by the Wellcome Trust (grants 220991/Z/20/Z and 203077/Z/16/Z), the Bill and Melinda Gates Foundation (INV-017547), and the Foreign Commonwealth and Development Office (FCDO) through the East Africa Research Fund (EARF/ITT/039) and is part of an integrated programme of SARS-CoV-2 sero-surveillance in Kenya led by KEMRI Wellcome Trust Research Programme. A.A. is funded by a DFID/MRC/NIHR/Wellcome Trust Joint Global Health Trials Award (MR/R006083/1), J.A.G.S. is funded by a Wellcome Trust Senior Research Fellowship (214320) and the NIHR Health Protection Research Unit in Immunisation, I.M.O.A. is funded by the United Kingdom’s Medical Research Council and Department For International Development through an African Research Leader Fellowship (MR/S005293/1) and by the NIHR-MPRU at UCL (grant 2268427 LSHTM). G.M.W. is supported by a fellowship from the Oak Foundation. C.N.A. is funded by the DELTAS Africa Initiative [DEL-15-003], and the Foreign, Commonwealth and Development Office and Wellcome (220985/Z/20/Z). S.U. is funded by DELTAS Africa Initiative [DEL-15-003], L.I.O.-O. is funded by a Wellcome Trust Intermediate Fellowship (107568/Z/15/Z). R.A is funded by National Institute for Health Research (NIHR) (project reference 17/63/82) using UK aid from the UK Government to support global health research.

The views expressed in this publication are those of the authors and not necessarily those of the funding agencies. The funders had no role in study design, data collection and analysis, decision to publish, or preparation of the manuscript.”

Response: Please include the amended statement below: “This project was funded by a commercial source, the Wellcome Trust (grants 220991/Z/20/Z and 203077/Z/16/Z). This does not alter our adherence to PLOS ONE policies on sharing data and materials. The funders had no role in study design, data collection and analysis, decision to publish, or preparation of the manuscript. Authors declare no other competing interests.”

Reviewer comments: 

Reviewer #1: 

- This is a succint report of seropositivity to SARS-CoV-2 in Kenya. Well executed and well presented. My major comment is that the infection with the virus, as evidenced by the presence of antibodies in blood samples, is considered the prevalence of the disease. Many patients were asymptomatic, in some the antibodies could be effective at fighting infection and thus preventing any symptoms. This comment is not directed specifically at the Authors, but refers to the general confusion of the disease with the infection by its causative agent. Infection does not mean disease (presence of disrupted organismal homeostasis). In the "Discussion" the Authors repeat details of the results unnecessarily. The last sentence of the Discussion (lines 319-320) is insightful. As a South African who lived through the years of the downfall of the apartheid, I found the use of the acronym ANC confusing (African National Congress) and the other acronyms interferring with understanding of the text. Thre is no list of abbreviations, but I would encourage Authors to avoid the use of acronyms in a publication on-line whose cost is not affected by the number of words, or characters, used in the text.

Response: We thank the reviewer for their comments, we agree the acronym maybe confusing, and we have removed it from the manuscript in this revision. 

Reviewer #2: 

- The authors of this manuscript, entitled “Sero-surveillance for IgG to SARS-CoV-2 at antenatal care clinics in three Kenyan referral hospitals: repeated cross-sectional surveys 2020-21”, presented a cross-sectional study aimed at assessing the transmission of SARS-CoV-2 in Kenya. The manuscript fits the aim of “PLOS ONE,” which publishes papers and research in the area of the natural sciences, medical research, engineering, and the related social sciences and humanities.The title, as well as the abstract, are informative; nevertheless, the study’s aim has not been clearly described and needs to be elucidated in more detail. Introduction requires a deeper contextualization of SARS-CoV-2 infection. Finally, the text should be written in a scientific style containing passive voice and 3rd person. Moreover, check the references list and the references cited in the text. Please review them for completeness and errors and note that according to PLOS ONE, the references must be cited in the Vancouver style.Therefore, I strongly recommend accepting the manuscript after minor revision.

Response: We thank the reviewer for their comments, we have stated the aim on line 89-90 of the manuscript “We aimed to determine the prevalence of antibodies against SARS-CoV-2 in mothers attending antenatal care at three referral hospitals in Kenya”. We have not changed ‘voice’ due to a stated preference for active voice on the submission guidelines. We have reformatted the references according to the editorial guidelines.

---

## [Editor Report · Decision Letter 1]

13 Sep 2022

Sero-surveillance for IgG to SARS-CoV-2 at antenatal care clinics in three Kenyan referral hospitals: repeated cross-sectional surveys 2020-21

PONE-D-22-06158R1

Dear Dr. Gallagher, 

We’re pleased to inform you that your manuscript has been judged scientifically suitable for publication and will be formally accepted for publication once it meets all outstanding technical requirements.

Kind regards,

Gheyath K. Nasrallah

Academic Editor

PLOS ONE
---

## [Editor Report · Acceptance letter]

5 Oct 2022

PONE-D-22-06158R1 

Sero-surveillance for IgG to SARS-CoV-2 at antenatal care clinics in three Kenyan referral hospitals: repeated cross-sectional surveys 2020-21 

Dear Dr. Gallagher:

I'm pleased to inform you that your manuscript has been deemed suitable for publication in PLOS ONE. Congratulations! Your manuscript is now with our production department. 

Kind regards, 

on behalf of

Dr. Gheyath K. Nasrallah 

Academic Editor

PLOS ONE